# Association of fish intake with menstrual pain: A cross-sectional study of the Japan Environment and Children's Study

Emi Yokoyama[1], Takashi Takeda[1,2]*, Zen Watanabe[1], Noriyuki Iwama[1,3], Michihiro Satoh[4], Takahisa Murakami[4], Kasumi Sakurai[5], Naomi Shiga[1], Nozomi Tatsuta[5], Masatoshi Saito[1,6], Masahito Tachibana[1], Takahiro Arima[5], Shinichi Kuriyama[3,5], Hirohito Metoki[3,4], Nobuo Yaegashi[1,3,5]

1 Department of Obstetrics and Gynecology, Tohoku University Graduate School of Medicine, Sendai, Miyagi, Japan, 2 Division of Women's Health, Research Institute of Traditional Asian Medicine, Kindai University, Osaka-Sayama, Osaka, Japan, 3 Tohoku Medical Megabank Organization, Tohoku University, Sendai, Miyagi, Japan, 4 Division of Public Health, Hygiene and epidemiology, Tohoku Medical and Pharmaceutical University Faculty of Medicine, Sendai, Miyagi, Japan, 5 Environment and Genome Research Center, Tohoku University Graduate School of Medicine, Sendai, Miyagi, Japan, 6 Department of Maternal and Fetal Therapeutics, Tohoku University Graduate School of Medicine, Sendai, Miyagi, Japan

* take@med.kindai.ac.jp

**Data Availability Statement:** Data are unsuitable for public deposition due to ethical restrictions and legal framework of Japan. It is prohibited by the Act on the Protection of Personal Information (Act

## Abstract

The relationship between fish eating habits and menstrual pain is unknown. Elucidating this relationship can inform dietary guidance for reproductive age women with menstrual pain. The aim of this study was to clarify the relationship between fish intake frequency/preference and menstrual pain. This cross-sectional study was conducted at the Miyagi Regional Center as an adjunct study of the Japan Environment and Children's Study, and 2060 eligible women (mean age, 31.9 years) participated. Fish intake frequency ("< 1 time/week," "1 time/week," "2–3 times/week," or "≥ 4 times/week"), preference ("like," "neutral," or "dislike"), and menstrual pain (no/mild or moderate-to-severe) were assessed at 1.5 years after the last delivery through self-administered questionnaires. The association between fish intake frequency/preference and prevalence of moderate-to-severe menstrual pain was evaluated through logistic regression analyses. Our results show that, compared with the "< 1 time/week" (38.0%) group, the "1 time/week" (26.9%), "2–3 times/week" (27.8%), and "≥ 4 times/week" (23.9%) groups showed a lower prevalence of moderate-to-severe menstrual pain ($p < 0.01$). The prevalence of moderate-to-severe menstrual pain was 27.7%, 27.6%, and 34.4% in the "like," "neutral," and "dislike" groups, respectively. Multivariate logistic regression showed that frequent fish intake was associated with a lower prevalence of moderate-to-severe menstrual pain ("1 time/week": odds ratio [OR] = 0.59; 95% confidence interval [CI], 0.41–0.86, "2–3 times/week": OR = 0.64; 95% CI, 0.45–0.90 and "≥ 4 times/week": OR = 0.52; 95% CI, 0.34–0.80; trend $p = 0.004$). Multivariate logistic regression showed no association between fish preference and moderate-to-severe menstrual pain ("dislike" vs "like": OR = 1.16; 95% CI, 0.78–1.73). There was a significant negative association between fish intake frequency and menstrual pain. It is suggested that fish intake can reduce or prevent menstrual pain.

No.57 of 30 May 2003, amendment on 9 September 2015) to publicly deposit the data containing personal information. Ethical Guidelines for Epidemiological Research enforced by the Japan Ministry of Education, Culture, Sports, Science and Technology and the Ministry of Health, Labour and Welfare also restricts the open sharing of the epidemiologic data. All inquiries about access to data should be sent to: jecs-en@nies.go.jp. The person responsible for handling enquiries sent to this e-mail address is Dr Shoji F. Nakayama, JECS Programme Office, National Institute for Environmental Studies.

**Funding:** The JECS was funded by the Ministry of the Environment, Japan. This research was partly supported by Grants-in-Aid for Scientific Research (No.16H05243; https://kaken.nii.ac.jp/ja/grant/KAKENHI-PROJECT-16H05243/ and No. 21K09508; https://kaken.nii.ac.jp/ja/grant/KAKENHI-PROJECT-21K09508/) from the Japan Society for the Promotion of Science. The funders had no role in the study design, data collection and analysis, decision to publish, or preparation of the manuscript. The findings and conclusions of this article are solely the responsibility of the authors and do not represent the official views of the Ministry of the Environment, Japan.

**Competing interests:** The authors have declared that no competing interests exist.

**Abbreviations:** ANOVA, Analysis of variance; BMI, Body mass index; CI, Confidence intervals; EPDS, Edinburgh Postnatal Depression Scale; JECS, Japan Environment and Children's Study; JPY, Japanese yen; OR, Odds ratio; PG, Prostaglandins; PUFA, Polyunsaturated fatty acids; SD, Standard deviation.

## Introduction

Dysmenorrhea, or menstrual pain, is the most common gynecological problem worldwide [1]. Menstrual pain usually lasts 24–48 hours from the beginning of menstruation [2]. The prevalence of dysmenorrhea varies widely (range, 15–84%) [3–5], and the highest prevalence is in adolescents [6]. Consequently, epidemiologic studies of dysmenorrhea are often limited to adolescents. However, dysmenorrhea also affects women of reproductive age, and can have a negative impact on many aspects of personal life, including family relationships, friendships, school and work performance, social life and recreational activities [7, 8]. Thus, given the significant impact of dysmenorrhea on productivity, it ultimately can have severe worldwide economic consequences [9, 10]. Therefore, research on dysmenorrhea across all age groups is needed.

The most widely accepted pathophysiologic mechanism of primary dysmenorrhea is the overproduction of uterine prostaglandins (PG) [10]. High menstrual fluid PGF2$\alpha$ levels were found in women with dysmenorrhea [11, 12]. Non-steroidal anti-inflammatory drugs can provide effective pain relief for women with dysmenorrhea [13]. However, serious side effects can occur after long-term treatment with these drugs [14]; therefore, dietary changes and supplements have received interest as alternative nonpharmacological medical approaches for dysmenorrhea.

Fish intake was recently reported to have a beneficial effect on systemic inflammation [15–17]. Fish consumption reduces the risk of coronary artery death [18] and carcinomas [19–21], and is negatively associated with postpartum depression [22]. Although there are many epidemiological studies on dysmenorrhea [3, 23], few studies have examined the relationship between fish intake and dysmenorrhea. Fish are a rich source of n-3 polyunsaturated fatty acids (PUFA), such as eicosapentaenoic acid, docosapentaenoic acid, and docosahexaenoic acid. The relationship between n-3 PUFA supplement intake and dysmenorrhea has yielded inconsistent results [24–27]. Thus, it is uncertain whether fish consumption is effective for dysmenorrhea as previous studies have not yielded consistent results due to small sample sizes.

The relationship between fish eating habits and menstrual pain is unknown. Elucidating this relationship is important because it can inform dietary guidance for reproductive age women with menstrual pain and may have a positive impact on many aspects of personal life. In addition, this relationship may be useful to healthy individuals in their daily food choices. Thus, this study aimed to assess the relationship between fish intake frequency/preference and menstrual pain among reproductive age women.

## Materials

### Study design

This cross-sectional study was conducted at the Miyagi Regional Center as an adjunct study of the Japan Environment and Children's Study (JECS). The JECS is a nationwide, government-funded birth cohort study, evaluating the impact of certain environmental factors on child health and development. The detailed study design has been previously described [28]. A total of 103,000 parent-child pairs were recruited from 15 areas in Japan from January 2011 to March 2014. Self-administered questionnaires were completed periodically, during pregnancy and after childbirth. A flowchart of the recruitment and exclusion process for women in the study is shown in Fig 1. At Miyagi Regional Center, 9,318 pregnant women participated in the main study and 3,793 agreed to participate in an adjunct study. Written informed consent was obtained from all participants. The adjunct study questionnaires were not sent to 103 women who were excluded due to various reasons. Questionnaires were sent to 3,690 women, and

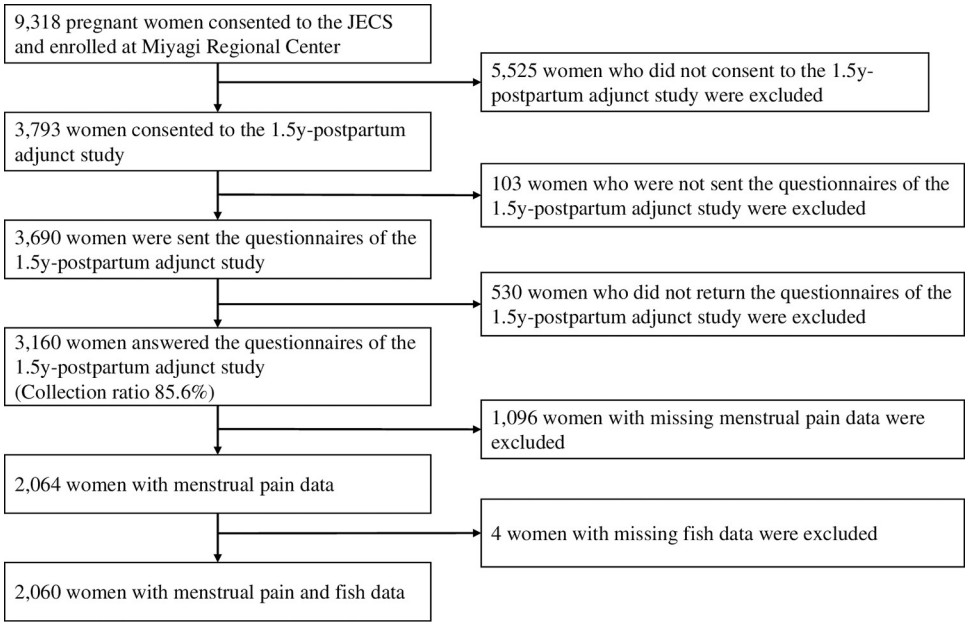

**Fig 1. Participant selection flowchart.**

3,160 women returned the questionnaires (collection ratio 85.6%). In addition, women with missing critical data were excluded, including 1,096 women with missing menstrual pain data. Furthermore, 4 women with missing data regarding fish consumptions/preferences were excluded, leaving 2,060 participants with data available for the study.

This study was approved by the Ethics Committee of Tohoku University School of Medicine (2021-1-187) and was therefore performed in accordance with the ethical standards laid down in the 1964 Declaration of Helsinki and its later amendments.

## Data collection

Self-administered questionnaires were administered at 12–16 weeks of gestation, 24–28 weeks of gestation, delivery, six months after delivery, and 1.5 years after delivery. We gathered data on demographic factors (physical, mental health, lifestyle, occupation, environmental exposure, habitation), and socioeconomic status, among other factors [29].

**Menstrual pain.** The questionnaire sent to participants at 1.5 years after delivery contained data on the characteristics of menstruation. Based on the experience of the preceding three months, participants were asked to answer the question, "What is the degree of pain during menstruation?". Participants rated their degree of menstrual pain based on their individual discretion because no clear criteria, such as impact on daily life, were set for the degree of menstrual pain. The degree of menstrual pain was classified into four categories: "painless," "mild," "moderate," and "severe" (most intolerable). To make the analysis easier to interpret, as reported in a previous study [30], the four categories of the severity of menstrual pain were collapsed into two: "painless to mild" and "moderate to severe".

**Fish intake frequency and preference.** Participants were also asked about their fish intake frequency and preference in a questionnaire at 1.5 years after delivery. Fish intake frequency was classified into four categories: "< 1 time/week" if participants do not eat fish at all or eat less than once a week, "1 time/week", "2–3 times/week", and "≥ 4 times/week" if participants eat fish 4–6 times a week or every day. Regarding fish preference, we asked participants to

answer the question, "Do you like eating fish?". Fish preferences were divided into three categories: "dislike" if participants disliked or slightly disliked fish, "neutral" if they did not like or dislike fish, and "like" if they liked or slightly liked fish. The classification of fish intake frequency and preference is based on the brief-type self-administered diet history questionnaire, and its validity and reliability have been reported [31, 32].

**Covariates.** Based on previous studies [3, 23, 33, 34], we selected baseline characteristics as covariates and categorized. S1 Table presents the details on the covariates used.

## Statistical analysis

Group differences in baseline characteristics were evaluated using the chi-square test. Continuous variables were compared by one-way analysis of varianve (ANOVA). We evaluated the prevalence of moderate-to-severe menstrual pain by fish intake frequency and preference and estimated the risk of moderate-to-severe menstrual pain. A logistic regression analysis and a multivariable logistic regression analysis were performed to calculate the odds ratios (OR) and 95% confidence intervals (CI). Model 1 was a simple model. Model 2 was adjusted for the following potential confounding factors: age, body mass index (BMI), smoking habit, passive smoking, alcohol intake, couple's education level, employment, family income, marital status, parity, fetal number, mode of delivery, obstetric complication, age at menarche, history of gynecological disease and mental illness, and postnatal depression. All the possible confounding factors were classified into categories as shown in Table 1, with missing values also categorized as Missing, and then used for adjustment. In the trend test, each category was evaluated as a continuous variable. In this study, missing covariates were included in multivariable logistic regression analyses as dummy variables. In addition, all baseline characteristics were stratified, and we evaluated the interaction between characteristics and fish intake frequency for the risk of moderate-to-severe menstrual pain.

Two-sided $p$ values less than 0.05 were considered statistically significant. All analyses were performed using SAS version 9.4 software (SAS Institute Inc., Cary, NC, USA).

## Results

Table 1 shows the baseline characteristics and the prevalence of menstrual pain, categorized according to fish intake frequency. When participants were stratified by the fish intake frequency, the average ages in the "< 1 time/week" and "≥ 4 times/week" groups were 30.5 and 32.3 years ($p < 0.01$). Other distributions in the "< 1 time/week" and "≥ 4 times/week" groups were as follows: participants' last education was university, 44.2% and 57.5% ($p = 0.02$), annual household incomes of < 4 million JPY, 55.8% and 36.2% ($p = 0.01$); primiparous women, 46.7% and 38.8% ($p = 0.01$); and a history of mental illness, 12.7% and 5.6% ($p = 0.04$), respectively. Moderate-to-severe menstrual pain was significantly more common in the "< 1 time/week" group (38.0%) than in the "1 time/week " (26.9%),"1 time/week " (27.8%), and "< 1 time/week " groups (23.9%) ($p < 0.001$).

S2 Table shows the baseline characteristics and the prevalence of menstrual pain, categorized according to fish preference. The "dislike" and "like" groups, respectively, comprised 53.7% and 47.3% of workers and 21.6% and 12.6% of women with postnatal depression. Unlike women who liked fish, women who disliked fish tended to be workers and have postpartum depression. The "like," "neutral," and "dislike" groups had moderate-to-severe menstrual pain (27.7%, 27.6%, and 34.4%, respectively).

Table 2 shows the results of the logistic regression analyses evaluating the association between fish intake frequency and the incidence of moderate-to-severe menstrual pain. Regarding fish intake frequency, in *Model 1*, the "1 time/week," "2–3 times/week," and "≥ 4

**Table 1. Baseline characteristics according to fish intake frequency.**

| | Fish intake frequency | | | | p value* |
|---|---|---|---|---|---|
| | <1 time/week | 1 time/week | 2–3 times/ week | ≥4 times/ week | |
| N(%) | 197(9.6) | 413(20.1) | 1182(57.4) | 268(13.0) | |
| Age (years) | | | | | |
| Mean (SD) | 30.5(5.0) | 31.9(5.0) | 32.1(5.0) | 32.3(5.0) | <0.001 |
| ≤24 | 27(13.7) | 26(6.3) | 73(6.2) | 17(6.3) | 0.003 |
| 25–29 | 56(28.4) | 108(26.2) | 293(24.8) | 55(20.5) | |
| 30–34 | 63(32.0) | 149(36.1) | 384(32.5) | 102(38.1) | |
| 35–39 | 36(18.3) | 88(21.3) | 326(27.6) | 65(24.3) | |
| ≥40 | 9(4.6) | 33(8.0) | 78(6.6) | 19(7.10 | |
| Missing | 6(3.1) | 9(2.2) | 28(2.4) | 10(3.7) | |
| BMI (kg/m$^2$) | | | | | |
| <18.5 | 28(14.2) | 41(9.9) | 130(11.0) | 32(11.9) | 0.67 |
| 18.5–24.9 | 132(67.0) | 287(69.5) | 806(68.2) | 176(65.7) | |
| ≥25 | 25(12.7) | 68(16.5) | 188(15.9) | 42(15.7) | |
| Missing | 12(6.1) | 17(4.1) | 8(4.9) | 18(6.7) | |
| Smoking habit | | | | | |
| Non-smoker | 163(82.7) | 359(86.9) | 1023(86.6) | 231(86.2) | 0.46 |
| Current smoker | 26(13.2) | 48(11.6) | 134(11.3) | 29(10.8) | |
| Missing | 8(4.1) | 6(1.5) | 25(2.1) | 8(3.0) | |
| Passive smoking | | | | | |
| Non-smoker | 104(52.8) | 217(52.5) | 639(54.1) | 145(54.1) | 0.58 |
| Current smoker | 81(41.1) | 183(44.3) | 505(42.7) | 113(42.2) | |
| Missing | 12(6.1) | 13(3.2) | 38(3.2) | 10(3.7) | |
| Alcohol intake | | | | | |
| None | 137(69.5) | 274(66.3) | 792(67.0) | 179(66.8) | 0.68 |
| Current drinker | 58(29.4) | 135(32.7) | 370(31.3) | 82(30.6) | |
| Missing | 2(1.0) | 4(1.0) | 20(1.7) | 7(2.6) | |
| Maternal educational level | | | | | |
| Junior high school | 8(4.1) | 27(6.5) | 50(4.2) | 9(3.4) | 0.02 |
| High school | 100(50.8) | 191(46.3) | 487(41.2) | 101(37.7) | |
| College | 87(44.2) | 188(45.5) | 632(53.5) | 154(57.5) | |
| Missing | 2(1.0) | 7(1.7) | 13(1.1) | 4(1.5) | |
| Paternal educational level | | | | | |
| Junior high school | 19(9.6) | 26(6.3) | 77(6.5) | 16(6.0) | 0.25 |
| High school | 92(46.7) | 230(55.7) | 598(50.6) | 133(49.6) | |
| College | 85(43.2) | 149(36.1) | 493(41.7) | 114(42.5) | |
| Missing | 1(0.5) | 8(1.9) | 14(1.2) | 5(1.9) | |
| Employment | | | | | |
| Homemaker | 104(52.8) | 195(47.2) | 549(46.5) | 116(43.3) | 0.25 |
| Worker | 79(40.1) | 200(48.4) | 576(48.7) | 137(51.1) | |
| Missing | 14(7.1) | 18(4.4) | 57(4.8) | 15(5.6) | |
| Family income (×10$^4$ JPY) | | | | | |
| ≤199 | 15(7.6) | 21(5.1) | 52(4.4) | 9(3.4) | 0.01 |
| 200–399 | 5(48.2) | 158(38.3) | 408(34.5) | 88(32.8) | |
| 400–599 | 38(19.3) | 113(27.4) | 349(29.5) | 80(29.9) | |
| ≥600 | 34(17.3) | 80(19.4) | 272(23.0) | 66(24.6) | |
| Missing | 15(7.6) | 41(9.9) | 101(8.5) | 25(9.3) | |

(*Continued*)

**Table 1.** (Continued)

| | Fish intake frequency | | | | |
|---|---|---|---|---|---|
| | <1 time/week | 1 time/week | 2–3 times/ week | ≥4 times/ week | p value* |
| N(%) | 197(9.6) | 413(20.1) | 1182(57.4) | 268(13.0) | |
| Marital status | | | | | |
| Married | 181(91.9) | 388(94.0) | 1135(96.0) | 254(94.8) | 0.07 |
| Others | 15(7.6) | 22(5.3) | 46(3.9) | 12(4.5) | |
| Missing | 1(0.5) | 3(0.7) | 1(0.1) | 2(0.8) | |
| Parity | | | | | |
| Primipara | 92(46.7) | 146(35.4) | 401(33.9) | 104(38.8) | 0.01 |
| Multipara | 102(51.8) | 265(64.2) | 761(64.4) | 159(59.3) | |
| Missing | 3(1.5) | 2(0.5) | 20(1.9) | 5(1.9) | |
| Fetal number | | | | | |
| Singleton | 195(99.0) | 409(99.0) | 1177(99.6) | 266(99.3) | 0.55 |
| Multiple | 2(1.0) | 4(1.0) | 5(0.4) | 2(0.8) | |
| Mode of delivery | | | | | |
| Transvaginal | 166(84.3) | 340(82.3) | 72(82.2) | 214(79.9) | 0.66 |
| Caesarean | 31(15.7) | 73(17.7) | 210(17.8) | 54(20.2) | |
| Obstetric complications | | | | | |
| None | 110(55.8) | 236(57.1) | 667(56.4) | 152(56.7) | 0.90 |
| Yes | 87(44.2) | 175(42.4) | 510(43.2) | 116(43.3) | |
| Missing | 0(0.0) | 2(0.5) | 5(0.4) | 0(0.0) | |
| Age at menarche | | | | | |
| ≤11 | 56(28.4) | 143(34.6) | 378(32.0) | 81(30.2) | 0.14 |
| 12–13 | 95(48.2) | 198(47.9) | 563(47.6) | 134(50.0) | |
| ≥14 | 39(19.8) | 62(15.0) | 222(18.8) | 42(15.7) | |
| Missing | 7(3.6) | 10(2.4) | 19(1.6) | 11(4.1) | |
| History og gynecological disease | | | | | |
| No | 184(93.4) | 89(94.2) | 1087(92.0) | 247(92.2) | 0.30 |
| Yes | 13(6.6) | 23(5.6) | 95(8.0) | 20(7.5) | |
| Missing | 0(0.0) | 1(0.2) | 0(0.0) | 1(0.4) | |
| History of mental illness | | | | | |
| No | 172(87.3) | 376(91.0) | 1093(92.5) | 252(94.0) | 0.04 |
| Yes | 25(12.7) | 36(8.7) | 89(7.5) | 15(5.6) | |
| Missing | 0(0.0) | 1(0.2) | 0(0.0) | 1(0.4) | |
| Postnatal depression (EPDS ≥9 points) | | | | | |
| No | 157(79.7) | 337(81.6) | 1008(85.3) | 232(88.6) | 0.18 |
| Yes | 33(16.8) | 68(16.5) | 151(12.8) | 30(11.2) | |
| Missing | 7(3.6) | 8(1.9) | 23(2.0) | 6(2.2) | |
| Menstrual pain | | | | | |
| No pain | 35(17.8) | 75(18.2) | 210(17.8) | 55(20.5) | <0.001 |
| Mild | 87(44.2) | 227(55.0) | 643(54.4) | 149(55.6) | |
| Moderate | 57(28.9) | 92(22.3) | 299(25.3) | 55(20.5) | |
| Severe | 18(9.1) | 19(4.6) | 30(2.5) | 9(3.4) | |

* Calculated using chi-square tests for categorical variables or a one-way ANOVA for continuous normally distributed variables.

SD, standard deviation; BMI, body mass index; JPY, Japanese yen; EPDS, Edinburgh Postnatal Depression Scale; ANOVA, analysis of variance

**Table 2. Results of logistic regression analyses evaluating the association between fish intake frequency or fish preference and menstrual pain severity.**

| | Fish intake frequency | | | | |
|---|---|---|---|---|---|
| | <1 time/ week | 1 time/ week | 2–3 times/ week | ≥4 times/ week | P for trend |
| | (n = 197) | (n = 413) | (n = 1,182) | (n = 268) | |
| | OR | OR | OR | OR | |
| | (95% CI) | (95% CI) | (95% CI) | (95% CI) | |
| Model 1 | 1.00 | 0.60 | 0.63 | 0.51 | <0.001 |
| | (ref) | (0.42–0.86) | (0.46–0.86) | (0.34–0.76) | |
| Model 2 | 1.00 | 0.59 | 0.64 | 0.52 | 0.02 |
| | (ref) | (0.41–0.86) | (0.45–0.90) | (0.34–0.80) | |

OR, odds ratio; CI, confidence interval

Model 1: A crude model

Model 2: A multivariate model adjusting for age, body mass index, smoking habit, passive smoking, alcohol intake, education level of couple, job, family income, marital status, parity, fetal number, mode of delivery, obstetric complication, age at menarche, history of gynecological disease, history of mental illness, and postnatal depression.

times/week" groups had a significantly lower risk of moderate-to-severe menstrual pain than the "< 1 time/week" group ("1 time/week": OR = 0.60; 95% CI, 0.42–0.86, "2–3 times/week": OR = 0.63; 95% CI, 0.46–0.96 and "≥ 4 times/week": OR = 0.51; 95% CI, 0.34–0.76). After adjusting for possible confounding factors (*Model 2*), the associations remained significant ("1 time/week": adjusted OR [aOR] = 0.59; 95% CI, 0.41–0.86, "2–3 times/week": aOR = 0.64; 95% CI, 0.45–0.90 and "≥ 4 times/week": aOR = 0.52; 95% CI, 0.34–0.80). Furthermore, a trend analysis using *Model 2* showed a statistically significant tendency for infrequent fish intake to increase the risk of moderate-to-severe menstrual pain (trend $p = 0.02$).

Table 3 shows the results of the logistic regression analyses evaluating the association between fish preference and the incidence of moderate-to-severe menstrual pain. Regarding fish preference, no significant associations were observed in both *Models 1* and *2*.

Stratified analysis by all baseline characteristics showed that smoking history and parity significantly influenced the association between fish intake frequency and moderate-to-severe menstrual pain. Fig 2 shows the OR of moderate-to-severe menstrual pain stratified by smoking habit and parity.

**Table 3. Results of logistic regression analyses evaluating the association between fish preference and menstrual pain severity.**

| | Fish preference | | | |
|---|---|---|---|---|
| | Like | Neutral | Dislike | P for trend |
| | (n = 1,593) | (n = 333) | (n = 134) | |
| | OR | OR | OR | |
| | (95% CI) | (95% CI) | (95% CI) | |
| Model 1 | 1.00 | 0.99 | 1.37 | 0.20 |
| | (ref) | (0.77–1.30) | (0.94–1.98) | |
| Model 2 | 1.00 | 0.95 | 1.14 | 0.72 |
| | (ref) | (0.72–1.24) | (0.78–1.73) | |

OR, odds ratio; CI, confidence interval

Model 1: A crude model

Model 2: A multivariate model adjusting for age, body mass index, smoking habit, passive smoking, alcohol intake, education level of couple, job, family income, marital status, parity, fetal number, mode of delivery, obstetric complication, age at menarche, history of gynecological disease, history of mental illness, and postnatal depression.

## OR (95% CI) for moderate-to-severe menstrual pain

**Fig 2. Odds ratio (OR) for moderate-to-severe menstrual pain stratified by smoking habit and parity.** □ Model 1: crude model ■ Model 2: multivariate model adjusting for age, body mass index, smoking habit, passive smoking, alcohol intake, education level of couple, job, family income, marital status, parity, fetal number, mode of delivery, obstetric complication, age at menarche, history of gynecological disease, history of mental illness, postnatal depression.

In non-smokers, there was an association of fish intake frequency with menstrual pain, while there was no association of fish intake frequency and menstrual pain in current smokers, showing a significant negative interaction (*p* for interaction = 0.02). Multiparous women showed a less statistically significant but negative tendency for an association of fish intake frequency with menstrual pain, and primiparas showed a stronger association between fish intake frequency and menstrual pain, with significant positive interactions (*p* for interaction = 0.05). All baseline characteristics, except the smoking history and parity, did not affect between the association fish intake frequency and risk of moderate-to-severe menstrual pain (*p* for interaction > 0.1). In addition, the basic characteristics according to smoking habit or parity are shown in S3 Table. S3 Table shows that, compared with non-smokers, current smokers were younger, overweight, passive smokers, poorly educated, had a low income, had obstetric complications, and had postnatal depression. Compared with multiparous women, primiparas

were younger, non-smokers, had fewer obstetric complications, and were more likely to have postnatal depression.

## Discussion

There was a significant negative association between fish intake frequency and menstrual pain. No association was observed between preference of fish and menstrual pain. In the present study, we adjusted for many covariates, and only few studies have been able to adjust for such a large number of covariates to examine the association between fish intake frequency and the risk of menstrual pain.

Grandi et al. [3] reported no association between fish consumption and menstrual pain, but we found an association after examining the fish intake frequency in more detail. Previous study [25] reporting that menstrual pain was correlated with low intake of n-3 PUFAs in fish, was limited by small population and social and environmental factors were not considered, but our study had a large population and was able to adjust for many covariates. Several randomized clinical trials [26, 35] have reported that fish oil is effective for menstrual pain, and our study supports this.

PG release is a pathogenetic factor in dysmenorrhea [6, 10]. A series of structural and biological PGs are formed from a series of different fatty acids. PGE2 and PGF2α, which are metabolites of n-6 fatty acids, are pro-inflammatory. The increased release of PGE2 and PGF2α, allegedly from cell disruption during endometrial sloughing, causes hypercontraction of the myometrium, resulting in ischemia and hypoxia of the uterine muscle, and ultimately, pain [10, 24]. Fish are a rich source of n-3 PUFAs, which inhibit the synthesis of endometrial PGF2α by competing with arachidonic acid (AA), a precursor of cyclooxygenase-2 [36]. N-3 PUFAs can also inhibit AA formation at the level of $\delta^6$-desaturase [37]. These effects of n-3 PUFAs may relieve menstrual pain.

In addition to n-3 PUFAs, fish are rich in nutrients, such as vitamin D and vitamin E, which may affect menstrual pain. Postnatal depression was reported to be more strongly associated with fish intake than n-3 PUFAs intake alone [38]. Vitamin D and the vitamin D receptor are involved in calcium homeostasis and different metabolic pathways as well as modulation of reproductive processes in humans [39]. The endometrium is a vitamin D target, and vitamin D receptor is expressed in the human uterus [40]. Vitamin D reduces the synthesis of PGs [41]. A single oral dose of vitamin D improved primary dysmenorrhea [42]. Vitamin E also improves menstrual pain. Vitamin E has an antioxidant effect, which reduces phospholipid peroxidation and inhibits the release of AA and its conversion to PGs [43, 44]. Therefore, it can play a significant role in relieving the severity of dysmenorrhea [43, 45]. On the contrary, there are studies that each nutrient alone is ineffective for alleviating menstrual pain [27, 46] or that a combination of nutrients enhances the effect against menstrual pain [26]. Fish may contain ω-3 fatty acids, vitamin D, and vitamin E, which may reduce menstrual pain.

There was a statistically significant negative interaction between "smoking habit" and "fish intake frequency" for moderate-to-severe menstrual pain. Some studies have suggested that nicotine, the major component in tobacco, could cause vasoconstriction, which can result in myometrial contraction due to hypoxia [1, 47]. Vasoconstriction reduces endometrial blood flow, causing menstrual pain. Smoking reduces the benefit of fish intake on menstrual pain.

There was a statistically significant positive interaction between being "primipara" and "fish intake frequency" for moderate-to-severe menstrual pain. In our study, primiparas were younger and more likely to have postnatal depression than multiparous women. Age and postnatal depression are associated with menstrual pain [48–50]; therefore, it is suggested that the primipara population may have been more susceptible to fish intake effects.

As reported in the Japanese dietary survey [51], fish intake among Japanese people is decreasing year by year, especially in young people. In this study, there was no difference in fish preference among different age groups, but the proportion of young people was high in the low fish intake group. This study population is representative of the real-world situation in Japan. Although it is known that fish are good for human health [15], several factors can influence the frequency of fish intake, such as price, supply, substitute goods, income, and taste. However, the reasons for lower fish intake among young people is not clear.

The participants in the JECS may be a health-conscious population, and fish preferences and intake frequencies may not always match because the participants may dislike, but still consume fish, for health reasons. In Japan, fish is more expensive than meat [51]; therefore, not all individuals who like fish can buy it. In Japan's food-oriented survey, "health-oriented," "simplified-oriented," and "economic-oriented" are ranked high [51]. It is speculated that fish preference and menstrual pain may not be related due to various factors.

There are several limitations to our study. First, because of its cross-sectional design, we were unable to clarify causality. Second, the sample size was relatively small; thus, our findings should be interpreted with caution. Third, this study assessed only fish intake frequency and may not accurately reflect fish consumption or type of fish; therefore, we could not assess the total intake of each nutrient, including n-3 fatty acids. On the other hand, a self-reported population with a high fish intake may have a higher n-3 PUFA bioavailability than one with a low fish intake [52]. This study used a self-administered questionnaire and may lack objective diagnosis. In addition, the influence of non-reported food items, caloric intake, or prenatal fish intake could not be excluded.

## Conclusion

There was a significant negative association between fish intake frequency and the risk of menstrual pain in Japan. The possible role of fish intake in alleviating menstrual pain is worth a closer examination using other study designs, such as longitudinal and/or intervention studies. It is suggested that fish intake can reduce or prevent menstrual pain.

## Supporting information

**S1 Table. Covariate settings and measurement times.**
(PDF)

**S2 Table. Baseline characteristics according to fish preference.**
(PDF)

**S3 Table. Baseline characteristics according to smoking habit or parity.**
(PDF)

## Acknowledgments

We would like to thank members of the Miyagi Regional Center of the Japan Environment & Children's Study Group and participants.

In addition, we would like to thank Editage (www.editage.jp) for English language editing.

## Author Contributions

**Conceptualization:** Emi Yokoyama, Takashi Takeda, Zen Watanabe.

**Data curation:** Emi Yokoyama, Zen Watanabe.

**Formal analysis:** Emi Yokoyama, Zen Watanabe, Hirohito Metoki.

**Funding acquisition:** Zen Watanabe, Hirohito Metoki.

**Investigation:** Emi Yokoyama, Zen Watanabe, Kasumi Sakurai, Nozomi Tatsuta, Takahiro Arima, Shinichi Kuriyama, Hirohito Metoki, Nobuo Yaegashi.

**Methodology:** Emi Yokoyama, Takashi Takeda, Zen Watanabe, Noriyuki Iwama, Michihiro Satoh, Takahisa Murakami, Naomi Shiga, Masatoshi Saito, Masahito Tachibana, Hirohito Metoki, Nobuo Yaegashi.

**Project administration:** Emi Yokoyama, Takashi Takeda, Zen Watanabe.

**Supervision:** Takashi Takeda, Zen Watanabe, Hirohito Metoki, Nobuo Yaegashi.

**Validation:** Emi Yokoyama, Takashi Takeda, Zen Watanabe.

**Visualization:** Emi Yokoyama.

**Writing – original draft:** Emi Yokoyama, Zen Watanabe.

**Writing – review & editing:** Emi Yokoyama, Takashi Takeda, Zen Watanabe, Noriyuki Iwama, Michihiro Satoh, Takahisa Murakami, Kasumi Sakurai, Naomi Shiga, Nozomi Tatsuta, Masatoshi Saito, Masahito Tachibana, Takahiro Arima, Shinichi Kuriyama, Hirohito Metoki, Nobuo Yaegashi.

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
