## [Decision Letter · Decision Letter 0]

2 Mar 2022

PONE-D-22-03678Association of fish intake with menstrual pain: an adjunct study of the Japan Environment and Children's StudyPLOS ONE

Dear Dr. Takeda,

Thank you for submitting your manuscript to PLOS ONE. After careful consideration, we feel that it has merit but does not fully meet PLOS ONE’s publication criteria as it currently stands. Therefore, we invite you to submit a revised version of the manuscript that addresses the points raised during the review process.

We look forward to receiving your revised manuscript.

Kind regards,

Antonio Simone Laganà, M.D., Ph.D.

Academic Editor

PLOS ONE

Journal Requirements:

Additional Editor Comments:

The reviewers have expressed positive comments regarding your article, raising only few concerns. Considering this point, I invite authors to perform the required minor revisions.

Reviewers' comments:

Reviewer's Responses to Questions

**Comments to the Author**

1. Is the manuscript technically sound, and do the data support the conclusions?

Reviewer #1: Yes

Reviewer #2: Yes

2. Has the statistical analysis been performed appropriately and rigorously? 

Reviewer #1: Yes

Reviewer #2: Yes

3. Have the authors made all data underlying the findings in their manuscript fully available?

Reviewer #1: Yes

Reviewer #2: Yes

4. Is the manuscript presented in an intelligible fashion and written in standard English?

Reviewer #1: Yes

Reviewer #2: Yes

5. Review Comments to the Author

Reviewer #1: Dear Editor

I read interestingly the manuscript entitled “Association of fish intake with menstrual pain: an adjunct study of the Japan Environment and Children’s Study”. The study seems conducted carefully.

Title

Authors can indicate that it is a cross-sectional study

- Please prepare a list of abbreviations at the beginning of the Meta DATA.

Abstracts

1) Abstract should be informative, background did not explain the question of this review and the answer which authors search for it

2) Keywords: are these keywords are Mesh terms? Word that serves as a keyword, as to the meaning of that condition must be a Mesh term

Introduction

The Introduction needs adjustments in order to answer these questions:

- What are the uncertainties and conflicts that underlie the hypotheticals?

- How important is the evidence of studies for the healthy individuals and patients?

- What is the focused clinical question your research will address?

Discussion

- The authors should list and shortly discuss the limitation of their study, for instance their limited number of participants and small sample size and also variation between type of fish that may affect the results

Reviewer #2: The fish intake before delivery or when she was little is unknown, and I think that this point is more important to investigate the association between fish consumption and dysmenorrhea. It must be described as a big limitation. However, it is epidemiologically interested the results and your discussion about the mechanism.

6. PLOS authors have the option to publish the peer review history of their article (what does this mean?). If published, this will include your full peer review and any attached files.

Reviewer #1: No

Reviewer #2: **Yes: **Satoshi Yoneda

---

## [Author Response · Author response to Decision Letter 0]

2 Apr 2022

Dear Editors and Reviewers

Thank you very much for reviewing our manuscript and for offering valuable advice. We have addressed your comments with point-by-point responses and revised the manuscript accordingly.

Responses to the Comments by the reviewer 1:

Thank you very much for taking the time to point out a number of critical points.

Title

Authors can indicate that it is a cross-sectional study

Response: Thank you for your kind suggestion. We have modified the "Title" to

“Association of fish intake with menstrual pain: A cross-sectional study of the Japan Environment and Children’s Study”

- Please prepare a list of abbreviations at the beginning of the Meta DATA.

Response: Thank you for your helpful recommendation. We have added a list of abbreviations before the "Abstract" section.

Abstracts

1) Abstract should be informative, background did not explain the question of this review and the answer which authors search for it

Response: Thank you very much for your invaluable comments. As you importantly pointed out, the background of the Abstract was not adequately described. Therefore, we have enriched the background as follows. 

“The relationship between fish eating habits and menstrual pain is unknown. Elucidating this relationship can inform dietary guidance for reproductive age women with menstrual pain.” (Page 4, Lines 36–38).

2) Keywords: are these keywords are Mesh terms? Word that serves as a keyword, as to the meaning of that condition must be a Mesh term

Response: Thank you for your helpful recommendation. We have modified the Keywords to Mesh terms.

“dysmenorrhea, menstrual pain, cross-sectional study, feeding behavior.” 

(On submission system)

Introduction

The Introduction needs adjustments in order to answer these questions:

- What are the uncertainties and conflicts that underlie the hypotheticals?

- What is the focused clinical question your research will address?

Response: Thank you very much for your invaluable comments. We apologize for the insufficient information about uncertainties in previous reports and clinical questions. The clinical question is whether fish consumption is effective for dysmenorrhea. As you have mentioned, we have supplemented the "Introduction" section.

“Thus, it is uncertain whether fish consumption is effective for dysmenorrhea as previous studies have not yielded consistent results due to small sample sizes.” (Pages 6–7, Lines 86–88)

- How important is the evidence of studies for the healthy individuals and patients?

Response: Thank you for pointing this out. As you mentioned, there was a lack of description of the importance of our study to healthy individuals and patients. Thus, I have added the following description in the Introduction section.

“Elucidating this relationship is informative because it can inform dietary guidance for reproductive age women with menstrual pain and may have a positive impact on many aspects of personal life. In addition, this relationship may be useful to healthy individuals in their daily food choices.” (Page 7, Lines 89–93)

Discussion

- The authors should list and shortly discuss the limitation of their study, for instance their limited number of participants and small sample size and also variation between type of fish that may affect the results

Response: Thank you very much for your important comments. As you pointed out, there are limitations regarding the limited number of patients and small sample size. Therefore, we have modified the Discussion section to include this limitation.

“Second, the sample size was relatively small; thus, our findings should be interpreted with caution.” (Page 24, Lines 324–325)

Also, as you mentioned, our questionnaire did not distinguish variation between type of fish, which may have affected the results. We have modified the text in the Discussion section, as follows: 

“Third, this study assessed only fish intake frequency and may not accurately reflect fish consumption or type of fish; therefore, we could not assess the total intake of each nutrient, including n-3 fatty acids.” (Page 24, Lines 325–327)

In addition, the strength of our study lies in the ability to adjust for many covariates. Thus, we modified as the introductory paragraph of the Discussion follows:

“In the present study, we adjusted for many covariates, and only few studies have been able to adjust for such a large number covariates to examine the associations between fish intake frequency and the risk of menstrual pain. To our knowledge, this is the first study performed with adequate meaningful power to examine associations between fish intake frequency and the risk of menstrual pain.” (Page 20, Lines 261–265)

Responses to the Comments by the reviewer 2:

1. The fish intake before delivery or when she was little is unknown, and I think that this point is more important to investigate the association between fish consumption and dysmenorrhea. It must be described as a big limitation. However, it is epidemiologically interested the results and your discussion about the mechanism.

Response: Thank you very much for your important comments. Our study does not have data on fish intake frequency before delivery. As you importantly pointed out, whether long-term fish intake is associated with dysmenorrhea or a short-term effect could not be demonstrated in this study and is a limitation of the present study. Therefore, we have modified the Discussion section to include this limitation.

“In addition, the influence of non-reported food items, caloric intake, or prenatal fish intake could not be excluded.” (Page 24, Lines 330–331).

Again, thank you for giving us the opportunity to strengthen our manuscript with your valuable comments and queries. We have worked hard to incorporate your feedback and hope that these revisions persuade you to accept our submission.

---

## [Decision Letter · Decision Letter 1]

13 May 2022

Association of fish intake with menstrual pain: A cross-sectional study of the Japan Environment and Children's Study

PONE-D-22-03678R1

Dear Dr. Takeda,

We’re pleased to inform you that your manuscript has been judged scientifically suitable for publication and will be formally accepted for publication once it meets all outstanding technical requirements.

Kind regards,

Antonio Simone Laganà, M.D., Ph.D.

Academic Editor

PLOS ONE

Additional Editor Comments (optional):

Authors performed the required corrections, which were positively evaluated by the reviewers. I am pleased to accept this paper for publication.

Reviewers' comments:

Reviewer's Responses to Questions

**Comments to the Author**

1. If the authors have adequately addressed your comments raised in a previous round of review and you feel that this manuscript is now acceptable for publication, you may indicate that here to bypass the “Comments to the Author” section, enter your conflict of interest statement in the “Confidential to Editor” section, and submit your "Accept" recommendation.

Reviewer #1: All comments have been addressed

2. Is the manuscript technically sound, and do the data support the conclusions?

Reviewer #1: Yes

3. Has the statistical analysis been performed appropriately and rigorously? 

Reviewer #1: N/A

4. Have the authors made all data underlying the findings in their manuscript fully available?

Reviewer #1: (No Response)

5. Is the manuscript presented in an intelligible fashion and written in standard English?

Reviewer #1: No

6. Review Comments to the Author

Reviewer #1: The revision of the manuscript was made, and I was satisfied with the response of the authors and do not have any more concerns.

7. PLOS authors have the option to publish the peer review history of their article (what does this mean?). If published, this will include your full peer review and any attached files.

Reviewer #1: No

---

## [Editor Report · Acceptance letter]

13 Jul 2022

PONE-D-22-03678R1 

Association of fish intake with menstrual pain: A cross-sectional study of the Japan Environment and Children’s Study 

Dear Dr. Takeda:

I'm pleased to inform you that your manuscript has been deemed suitable for publication in PLOS ONE. Congratulations! Your manuscript is now with our production department. 

Kind regards, 

on behalf of

Dr. Antonio Simone Laganà 

Academic Editor

PLOS ONE